# Putting Indigenous Cultures and Indigenous Knowledges Front and Centre to Clinical Practice: Katherine Hospital Case Example

**DOI:** 10.3390/ijerph21010003

**Published:** 2023-12-20

**Authors:** Carmen Parter, Josephine Gwynn, Shawn Wilson, John C. Skinner, Elizabeth Rix, Donna Hartz

**Affiliations:** 1Djurali Centre for Aboriginal & Torres Strait Islander Education and Health Research, Heart Research Institute, Newtown, NSW 2042, Australia; john.skinner@hri.org.au; 2Charles Perkins Centre, Faculty of Health Sciences, University of Sydney, Sydney, NSW 2006, Australia; josephine.gwynn@sydney.edu.au; 3Department of Community, Culture and Global Studies, University of British Columbia, Vancouver, BC V6T 1Z3, Canada; shawn.wilson@ubc.ca; 4Adelaide Nursing School, University of Adelaide, Adelaide, SA 5005, Australia; 5School of Nursing and Midwifery, Western Sydney University, Penrith, NSW 2751, Australia; d.hartz@westernsydney.edu.au

**Keywords:** Indigenous cultures, Indigenous knowledges, public healthcare policy, allies, accomplices, case example, cultural competency, cultural safety

## Abstract

The inclusion of Indigenous cultures, known as the cultural determinants of health, in healthcare policy and health professional education accreditation and registration requirements, is increasingly being recognised as imperative for improving the appalling health and well-being of Indigenous Australians. These inclusions are a strengths-based response to tackling the inequities in Indigenous Australians’ health relative to the general population. However, conceptualising the cultural determinants of health in healthcare practice has its contextual challenges, and gaps in implementation evidence are apparent. In this paper, we provide a case example, namely the Katherine Hospital, of how healthcare services can implement the cultural determinants of health into clinical practice. However, to be effective, health professionals must concede that Australia’s Indigenous peoples’ knowledges involving cultural ways of being, knowing and doing must co-exist with western and biomedical knowledges of health practice. We use the Katherine Hospital ABC Radio National Background Briefing interview, which was mentioned by two research participants in a 2020 study, as an example of good practice that we can learn from. Additionally, the six Aboriginal and Torres Strait Islander Health actions contained in the 2nd Edition of the Australian National Safety and Quality Health Service Standards provide governance and accountability examples of how to enable Indigenous people’s cultures and their knowledges in the provision of services. The role of non-Indigenous clinical allies and accomplices is imperative when embedding and enacting Indigenous Australians’ cultures in service systems of health. When Indigenous Peoples access mainstream hospitals, deep self-reflection by allies and accomplices is necessary to enable safe, quality care, and treatment that is culturally safe and free from racism. Doing so can increase cultural responsiveness free of racism, thereby reducing the inherent power imbalances embedded within mainstream health services.

## 1. Introduction

In Australia, the cultures of the Aboriginal and Torres Strait Islander people (respectfully known as Indigenous) have become prominent and central in the decisions and actions of governments (i.e., public policies) that aim to improve their health and well-being [1,2,3,4]. However, challenges remain with an apparent gap in evidence concerning how to implement Indigenous peoples’ cultures, known as the cultural determinants of health [5,6,7].

Drawing on new evidence, we provide a case example that was mentioned in a 2020 study [6] regarding how healthcare providers and services can permanently support the implementation of the cultural determinants of health into organizational and clinical practice.

The Australian Broadcasting Corporation (ABC) Radio National Background Briefing interview produced by Hagar Cohen about Aboriginal people taking their own leave at a hospital in a Territory of Australia—the Katherine Hospital in Northern Territory—was mentioned by two study participants. One study participant was so impressed by the interview that they said:

“the Katherine Hospital … a wonderful 40-min program on radio national … I make all my guys [i.e., staff] listen to that 40 min … cause it’s really, important thing … because it touches on so many issues … it’s definitely … a good case study”(P6, personal communication, June 2018)

Background Briefing is a weekly program exploring diverse social and political issues from across Australia; it is broadcast nationally on the ABC, which is Australia’s government-funded radio and television national broadcasting network.

In presenting this case example, an Indigenous research methodology of storying [8] is applied; it is also written in [6]. More importantly, we apply storytelling in this paper, which is a form of conversational style of storytelling and vignettes, while drawing out the lessons we learned from the Katherine Hospital as they tackled ‘taking their own leave’ (to be discussed below). We use the Katherine Hospital radio interview while weaving research participant yarns and stories in the material that is presented in this paper [6]. Yarning is a valid and rigorous Indigenous research method of storytelling [9,10,11]. In this paper, the lessons that are learnt from the case example are told through such methods and are further legitimatised by the Indigenous authors of this paper.

Three of the six co-authors of this paper are Indigenous scholars and thought leaders. The first author and another are Australian Indigenous academics with decades of expertise researching and publishing their work on the need to address systemic and individual racism and provide culturally safer healthcare services. The third co-author is an Indigenous scholar from Canada who is a global leader in Indigenous research methodologies. These scholars have an obligation to decolonize research in a way that centres Indigenous peoples’ knowledges and their ways of being, knowing, and doing, while producing new and original knowledges [12]. The three non-Indigenous co-authors are all committed allies or accomplices who have also researched and published work on racism and the barriers and enablers of Indigenous health in Australia and beyond.

### 1.1. How to Implement Culture?

A 2020 study examined how Indigenous culture can be implemented once incorporated into a public policy instrument such as Australia’s National Aboriginal and Torres Strait Islander Health Plan 2013–2023 (the Health Plan). Indigenous culture is central to the Health Plan’s policy framework [5].

In 2018, this study conducted nineteen individual yarning sessions, a conversational method involving storytelling, and a workshop with research participants who were active participants, known as policy actors, during the development of the Plan. These actors were drawn from the Indigenous policy sub-system of Australia’s large, complex, and political public health policy environment. They told their stories about the issues and challenges and provided examples of implementing Indigenous people’s cultures [6]. Their recorded stories were transcribed and uploaded into a data software management tool, NVivo, which were inductively coded thematically into a set of themes [6]. Implementing Indigenous cultures is about permanently embedding the cultural beliefs, practices, and knowledges held by Indigenous Australians [13]. This is a necessity if we are to transform government services [14]. Equally important is providing the means to enable the voices of Indigenous Australians to self-determine their health needs and care for their communities [5].

Further reflections of policy actors about the inclusion of Indigenous culture in the Health Plan are consistent with calls for culture to be at the centre of change as ‘an enabler of health and wellbeing’, as identified in the 2017 Australian Government Report *My Life My Lead: Opportunities for strengthening approaches to the social determinants and cultural determinants of Indigenous health* ([4]. As stated by a policy actor (notated throughout with ‘P’) who worked in government:

“I did pick up early on that without culture being central to the plan, we wouldn’t get terribly far … our Indigenous partners said … culture needs to be central, to this”.(P3, personal communication, June 2018)

Connection to culture has a strong relationship to good health and wellbeing [15]. Several research participants reiterated this, with one saying:

“if we’ve got a good connection to language, we’ve got good connection to your culture, ability to practice culture, access to land, and all those sorts of things, they’re strengths that add to wellbeing, and are important to Aboriginal health”.(P2, personal communication, June 2018)

While another explained:

“culture is a being. It’s not a thing; it’s not a commodity. It’s something within that defines who we are and our identity and what that means”.(P7, personal communication, July 2018)

However, another observed that:

“culture, as a concept, can be a little bit slippery to understand”.(P13, personal communication, September 2018)

### 1.2. The Six Cultural Domains of Indigenous Cultures—The Determinants of Health

Six cultural domains of the determinants of health have been identified concerning the cultures of Indigenous peoples globally: Connection to Country, involving spiritual connection; living on Country; land rights; autonomy; and caring for the Country, are all relevant [16]. Cultural beliefs and knowledge, including spiritual and religious beliefs, traditional knowledge, traditional healing, and knowledge transmission and continuity are vital to health and well-being. Language and its relevance to health and education are critical. Family, kinship, and community are essential. Expressions of culture and cultural identity, traditional practices, art and music, community practices, and sports are seen as necessary. Indigenous people’s leading and determining their destiny concerning cultural safety and well-being is imperative [16]. Aboriginal worldviews consider that a connection to Country encompasses far more than merely the physical. Country is interwoven with belonging and ways of believing, passed on from generation to generation through storytelling and yarning. Connection or reconnection to Country inherently embodies health and healing for Indigenous people and is a major cultural determinant of health [17].

These domains are now embedded into the Health Plan’s second Implementation Plan as determinants of health [18,19]. However, conceptualising culture into practice has its difficulties, with one study participant saying:

“the challenges of how to understand [culture] and then how to embed it … remains the critical issue”(P8, personal communication, July 2018)

and another asking:

“how do we build on the cultural sort of side of it? Is a really important question. So, what does that mean for our service system?”.(P11, personal communication, September 2018)

While another proclaimed:

“we can’t rely on cultural safety or cultural awareness training to support the implementation of culture”.(P6, personal communication, June 2018)

Cultural respect, cultural competency or cultural safety concepts and ways of praxis relate to expected attributes of how systems, organisations or individuals behave culturally when Indigenous Australians access healthcare (Personal communication, 26 September 2018).

Moreover, should healthcare providers and policymakers be expected to understand an Indigenous worldview? Several research participants confirmed that those developing programs and policies or providing services must understand Indigenous Australian’s cultures and worldviews of ways of being, knowing and doing (Personal communication, June–December 2018). Although, as one participant said,

“what’s important to recognize is that, is not to get, stereotypical about what culture means because it does mean different things to different people from practicing songs and dances and stories to also, culture of looking after the Country, respecting Country, having a relationship with Country, which are all important. So, it does differ around the nation, as to what it means”(P15, personal communication, September 2018)

Understanding Indigenous cultures and an Indigenous worldview is now a prerequisite of one of the six new Aboriginal and Torres Strait Islander health actions contained in the 2nd Edition of the National Safety and Quality Health Service Standards, which came into effect in 2019. These standards provide a nationally consistent statement of the level of care consumers can expect from health service organisations in Australia, especially Indigenous Australians. Specifically, health services are expected to provide a welcoming and an Indigenous-friendly environment that “recognises the importance of Indigenous Australia’s cultural beliefs and practices” [1,2]).

Importantly, the knowledges held by Indigenous Australians must be paramount, and its innate relationship to Indigenous cultures must be conceded. Innovative solutions are created when Indigenous people and their knowledges, cultures, and worldviews are valued [20]. While the space where many knowledges and cultures meet are highly contestable, an appreciation of working at the cultural interface is imperative [6,20,21,22,23].

## 2. The Case Example—Katherine Hospital

### 2.1. Background

The Katherine Hospital is approximately 320 kilometres southwest of Darwin (see Figure 1), the capital city of the Australian Northern Territory (NT) [23]. Katherine Hospital is a district public hospital servicing the Katherine Region in the NT. It services an area of 336,674 km^2^, with 85% percent of its patients being Aboriginal people, many from some of the most remote communities in Australia. It is operated by the NT Government Department of Health [24]. The Katherine region has a population of 20,000 (see Figure 2.) [23,24,25].

Communities continue to practice their cultural beliefs and knowledges, ceremonies, traditional medicines, and healing practices [26].

The Katherine Hospital 40-minutes radio national program examined a hospital in crisis, including how management and clinicians turned such a crisis around to then …” become rank[ed] as among Australia’s best for its relationship with Indigenous patients” … [26]. The Katherine Hospital radio interview highlighted several ways clinicians can incorporate Indigenous cultures, i.e., the determinants of health, into healthcare practice. We share these insights complemented by research participants’ stories from the 2020 study [6].

### 2.2. The Katherine Hospital’s Story—Taking Their Own Leave Rates

In 2010, Katherine Hospital’s medical model of care was untenable, representing the antithesis of Aboriginal culture and the holism of health [27]. Aboriginal patients were taking their own leave at alarming rates. The term ‘taking their own leave’ refers to when an admitted or non-admitted patient leaves a hospital or healthcare setting before their treating physician and healthcare team has authorised discharge [28]. In 2011, one in four Aboriginal patients took their own leave from Katherine Hospital “because they didn’t feel the hospital cared for them and they didn’t trust the doctors …or their treatment was inadequate” [26,29].

From June 2013 to June 2015, NT Aboriginal patients taking their own leave were 9.2 times the rate compared to non-Indigenous patients [30]. Nationally, it was 7.1 times the rate of non-Indigenous patients [30] and was highest in rural and remote areas [31], with a greater risk of worsening complications, even death, on return to a hospital [32].

Reasons for Indigenous Australians taking their own leave are complex, and include racism, inadequate or absence of cultural safety, and mistrust of the health system resulting from historical and ongoing systemic barriers [31]. Communication and language barriers, not understanding treatment and care [31], conflicting social and cultural obligations, isolation, and loneliness [31] are also significant contributors to taking one’s own leave.

### 2.3. What Katherine Hospital’s Management and Clinicians Did?

Katherine Hospital implemented a range of strategies to address the high rates of Aboriginal people taking their own leave, including providing additional support to junior clinicians [26].

Specifically, consideration was given to cultural factors, as was the case for a Jaowyn Elder who arrived with acute kidney failure. Without dialysis at a tertiary institution such as the Darwin Hospital, some 300 km away, she was unlikely to survive the weekend [26]. As a hospital clinician said, while:

“going to Darwin might save her life … Darwin hospital is too far from her Country and family”[29]

The Jaowyn Elder’s spiritual connection to Country and family became paramount and limited any immediate transfer to a tertiary institution like the Darwin Hospital. However, the service system often does not understand the importance of an Aboriginal Elder’s connection to their Country and family. For Indigenous Australians, ‘Country’ is far more than a reference to land or a home; it is a unique connection to land, like a family bond. ‘Country’ has inherent associations with spirituality, tradition, and ownership as the foundation of social structures, systems, and cultural practices entwined with the land. This relationship is vital for identity, health, and wellbeing [33].

As one study participant said:

“it can be a real battle and a fight … just getting the system to understand the importance of being on Country, if you’re gonna pass away. The importance of being treated closer to home and all those sorts of things. How can we improve that for people? So, it basically means they have to leave Country and I guess the outcomes aren’t good because people quite often just stop halfway, they don’t want to be removed from Country, they don’t wanna not have their family around them. That whole patient journey is really critical because of the vast distance between regional centers and [tertiary] hospital”.(P1, personal communication, June 2018)

While another non-Indigenous study participant expressed:

“I absolutely understand the connection to land, there is this deep, deep and ancient attachment to land. But how do I actually turn that into some words that are around, you know, what does this mean in a practical sense?”.(P3, personal communication, June 2018)

A hospital clinician (on the radio program) went on to say:

“so, the doctors are about to negotiate with her … it will be complex and requiring great knowledge in understanding about her culture and wishes … We will have to have a conversation … and it be good to have that with all the family”

Who then added:

“so the ED staff will be in the process of getting the family in to have a family discussion … she needs a big boss doctor to have that conversation with her and her family, and the family will know who the boss doctors are … and they will expect to have the boss doctor to talk to them”.[29]

Talking to family and not an individual, as expected in western non-Indigenous nuclear families, meant doctors acquired an understanding of the broader family structures and relationships in Aboriginal societies. Understanding the complexities and sensitivities of Aboriginal kinship relationships, particularly relational taboos, must be paramount. Neglecting such understanding compromises care [26]. As one study participant said:

“When you go to practice … you might not be aware [of] kinship relationships, of all these relationships … Some places these are very strong … brother sister … in some places you can’t … young men can’t talk to their sisters … so avoidance relationships … unless you understand that they’re operating … you’re going to end up doing something very silly”.(P9, personal communication, August 2018)

Adding:

“if you don’t have any cultural background or an education in what sort of cultural imperatives might be … when you go to practice, those things will be a challenge because you might be aware of them necessarily, you might not be aware”(P9, personal communication, August 2018)

who went on to say:

“[I] spent a lot of time where I went to work in the [de-identified] Country. I went and learned language so I could understand. I went and did some fairly, basic anthropology just so I could understand relationships, in the kinship system”(P9, personal communication, August 2018)

Concluding that:

“if you understand that that’s how the community sees itself, then it becomes easier to develop policy around how you practice healthcare”(P9, personal communication, August 2018)

Understanding cultural imperatives became apparent when treatment plans at Katherine Hospital began to integrate bush medicines and traditional healers into patient care. As a hospital clinician said:

“they want to put the bush medicine cream on them … I’m always happy for them to do whatever and sort of incorporate that in our treatment plan as well. So, this is what we’ll do with white man medicine, but we will also add … you know, we can do bush medicine as well for you if you bring it in”[29]

Further adding:

“I think in the meantime, we try to be relatively respectful in terms of bush medicine and black magic and witch doctors and things, and encourage it if they’d like to have that as part of their treatment, then certainly facilitate it”[29]

The term ‘black magic and witch doctors’ is used colloquially when describing the need for increased recognition and inclusion of diverse health beliefs and practices.

One study participant exclaimed:

“Mum uses something called [de-identified] … its properties are more sort of antiseptic, but you can also like drink it, and it will clear up any wounds really quickly, and if you’ve got a cough … a really, powerful bush medicine,… so how could we, … we should be tapping into that and actually, revitalizing that whole area of knowledge”.(P11, personal communication, September 2019)

The incorporation of bush medicines as an integral part of an Aboriginal patient’s care and inclusion of such treatment into their management plan is an example of implementing or embedding the cultural beliefs, knowledges, and practices of Aboriginal people into clinical care.

Additionally, understanding that the dialects spoken by Aboriginal people require interpreter services to minimise communication barriers was also considered. As one hospital clinician expressed:

“English isn’t their first language. It may not even be their second or third either. If there’s any doubt, we get interpreters in, or even on the phone”.[29]

A study participant reinforced:

“so, when trying to get someone to talk about their symptoms, unless they’re asking the right questions and drawing it out and the then patient needs someone who can interpret stuff a little bit and make them feel comfortable because a lot of people don’t want to talk about things … I think there’s too many opportunities for errors then. You need someone who can sort of do that sort of interpretation … cause language is still such a big barrier”(P11, personal communication, September 2018)

Importantly, metaphors were often used as the dialect had no word(s) to explain certain conditions like cancer. In the radio program, one hospital clinician described a fungal infection, for example as:

“You know like mushrooms growing”.[26]

Diabetes described as:

“sticky red blood that clogs up the pipes and causes blockage … how a medication will help pull the sugar out of the blood”.[29]

Critically, Katherine Hospital management team and clinicians also understood that:

“the responsibility to understand Aboriginal culture and worldviews became the business of everyone and not simply left to Aboriginal staff such as the Aboriginal Hospital Liaison Officer”.[29]

However, as stressed by a study participant:

“colonization has had an impact on culture and language too, across the nation… and, it’s so critically important for a lot of the policymakers and the influencers to gain that understanding as well”(P15, personal communication, September 2018)

This may enable policymakers, service providers and clinicians to become effective allies and accomplices in enabling, embedding, and enacting Indigenous cultures into policies, programs, and services [5].

The positive outcomes in this Katherine Hospital case example would not have been possible without clinicians practising as effective allies and accomplices to Indigenous Australians and their families when presenting for treatment and care.

### 2.4. Allies and Accomplice’s Role—Supporting Integration of Indigenous People’s Culture into Clinical Practice and Services

Working as a genuine non-Indigenous ally or accomplice to Aboriginal people in healthcare requires a critically reflective, culturally sensitive, and focused approach. When non-Indigenous healthcare professionals gain experience and knowledge about working in a culturally safe and capable way that is free of racism, they begin to grasp what being a true accomplice or ally looks like in their everyday practice [34]

Allies promote Indigenous voices above their own and may call out blatant instances of racism; however, they can struggle to identify institutional and individual micro-aggressions that embed racism and disrespect into healthcare in Australia. Accomplices, however, are prepared to stand up alongside Indigenous colleagues and clients, knowing when and how to step back to enable their Indigenous colleagues’ authority and the right to decide and define what action is required. Accomplices use their voices to stand up and call out injustices, racism, and micro-aggressions in healthcare environments, pushing back against the western/white dominance of healthcare delivery [35,36].

Definitions and understandings of the roles of allies and accomplices have expanded and deepened since the recent emergence of the BlackLivesMatter movement in the USA [37]. Anecdotally, the terms ally and accomplice are reflective of the nuances of language that underpin these terms. Ally is a word with strong links to political positioning in times of war and conflict; for example, Australia has been a strategic ally in the western Alliance since World Wars I and II. Accomplices are generally understood to be assistants to those charged with a criminal offence. There has been a default deficit attitude (since colonization) of non-Indigenous/white society towards Indigenous Peoples and their culture, which has driven systemic over-incarceration of Indigenous Peoples in Australia. This deficit lens often equates Indigenous Peoples with criminality. Accomplice can, therefore, be an appropriate and accurate term for those who are genuinely committed to Indigenous Peoples’ empowerment, self-determination, and sovereignty.

Active accomplices have developed strong positive therapeutic relationships with Indigenous People, understanding that connection to Country and culture are crucial to health and well-being. Accomplices promote Indigenous patients, Elders, families, and communities as the ‘experts’ in Indigenous health and wellbeing, holding relational accountability for their words and actions [9,38]. They understand that relationships, Country, and culture are at the heart of Indigenous worldviews, and they respect and use culturally shaped communication styles, such as yarning and storytelling, to build trust and rapport in clinical settings [39].

As members of the dominant cultural group in healthcare, healthcare professionals can only become genuine and active allies and accomplices by using critical, deep self-reflection that acknowledges unconscious biases, assumptions, and attitudes. Aiming to reduce inevitable power imbalances in health care is at the core of the cultural safety framework, which is now required within everyday clinical practice for healthcare workers in Australia that must be free of racism [40]. Allies and accomplices value the unique and vital role of Aboriginal Health Workers/Practitioners within hospitals and mainstream healthcare services in creating culturally safer services [41]. Allies and accomplices can also reduce the burden placed on Aboriginal Liaison Officers (ALOs) within hospitals, who are frequently called on to advise and support non-Indigenous staff working with their people. These demands on an ALO’s time by non-Indigenous workers can create untenable workloads, leading to the ALO feeling overwhelmed and burnt out with disturbing frequency [42]. Having allies and accomplices in hospitals can therefore significantly reduce the ALOs workload, leaving them free to perform their primary role, providing culturally safe support, comfort, and guidance to Aboriginal people as they navigate the health system while in hospital.

Effective allies and accomplices know ‘their place’; that is, they know how to respectfully occupy the space between the Indigenous and non-Indigenous worlds within healthcare settings. They know when to ‘step up’ and speak out (e.g., calling out systemic or individual racism), when to ‘walk alongside’ Indigenous patients and colleagues (e.g., decolonising healthcare practice) and when to ‘step back’ (e.g., enabling Indigenous clients and colleague’s leadership, truth telling, self-determination, and governance).

As evidenced by the radio program and the voices of study participants, non-Indigenous staff working with Indigenous People at the Katherine Hospital clearly acted as effective allies. They have contributed to turning a previously culturally unsafe environment into a more culturally comfortable and safe environment for local Aboriginal People to access healthcare. Further, they have increased respect for and have developed an understanding of Aboriginal people’s strong connection to Country, including those kinship relations of cultural ways of being, knowing, and doing. Additionally, the inclusion of the cultural determinants of health into clinical practice supports the rights of Indigenous Australians [43] to practice their cultural beliefs, knowledges, and traditional healthcare practices that create culturally safe, clinically responsive care that is free of racism [18].

## 3. Discussion

Culture has been exhibited as core to the National Aboriginal Health plan, i.e., the cultural determinants of health. Translating such policy into practice can be supported through service delivery examples, such as the Katherine Hospital, whereby embedding culture operationally influences better service delivery and outcomes. Katherine Hospital introduced a range of strategies to reduce the rates of Aboriginal people taking their own leave. Indigenous cultural beliefs and practices were taken into consideration when providing healthcare. Governing bodies of services like hospitals must take full responsibility for ensuring Indigenous knowledges and Indigenous cultures are enacted when providing services. Also, they must ensure that these strategies are built into an organisation’s planning, resource allocation, and data monitoring and accountability systems. Furthermore, their workplace must be culturally competent, and their workforce must be culturally safe and free from racism.

There is increasing recognition of the *cultural* determinants of health as being of primary importance in health services policy and delivery for all Indigenous Peoples [7,44]. The six cultural domains of Indigenous cultures—as listed under Section 1.2 above—also known as the determinants of health, are indeed crucial. Without the inclusion of these determinants of health within health policy and practice, Indigenous Peoples’ health and well-being will always be negatively impacted.

The work of committed non-Indigenous accomplices is important in ‘value adding’ to health workplace cultural safety and competence, and ideally, accomplices can informally increase culturally shaped treatment and care as they lead their colleagues by demonstrating how that can be translated into everyday practice. Allies and accomplices can lead the way in increasing respect for and understanding of the necessity of culturally responsive care to create accessible and acceptable healthcare for Aboriginal people. They can also model strategies to assist their colleagues in finding practical ways to decolonise their everyday practice within healthcare institutions, especially hospitals [45]. For example: engaging in culturally comfortable communication styles, such as yarning and storytelling, and building connection through self-disclosure; engaging with Indigenous media outlets and publications; and going to Indigenous cultural events [35].

Unfortunately, the deliberately hidden history of the brutality of colonization (by the dominant western/white cultural group) prevails, and subsequent embedded systemic racism remains a major barrier to implementing the cultural determinants of health, and culturally safe and responsive care, including two-way understanding and building positive therapeutic relationships within hospitals [38,46]. In mid-October 2023, Indigenous Australians observed the brutality of colonization when our nation voted in a referendum on a proposal to recognize Indigenous Peoples in Australia’s constitution and allow an Indigenous Voice, to inform and provide advice regarding policy development that directly impacted their People. Following a protracted, divisive campaign of misinformation and confusing rhetoric, this proposal was firmly rejected by a majority of voters, creating further division, and widening the cultural chasm that exists between Indigenous and white Australians [47]. Allies and accomplices can play a role in reducing this chasm by urging institutions to prioritize access to Indigenous-led education strategies for all clinical staff, and lead by example in decolonizing clinical practice, communication, and policy, especially while implementing the cultural determinants of health [35,48].

Importantly, Indigenous consumers, patients, carers, and Indigenous staff must be involved in assessing whether services are culturally competent and culturally safe [40]. Finally, working in equal partnership with Indigenous Australians is imperative, particularly as culture is now a determinant of health [18]. These good practice elements are espoused in the 2nd Edition of the National Safety and Quality Health Service Standards [1].

In closing, Indigenous communities have been drawing on case examples in the form of storytelling and metaphor in their efforts to educate mainstream health systems about the crucial roles of cultural beliefs, knowledges, ceremonies, and traditional medicines since colonization. More research in this area is needed to expand non-Indigenous policy actors’ and clinicians’ knowledge and awareness of the critical role that culture and Indigenous ways of knowing, being, and doing, play out, in health and well-being. There is also a lack of research on the roles that non-Indigenous allies and accomplices can play in supporting Indigenous people in their quest for accessible, acceptable, culturally competent and safer healthcare services, in which self-determination is enhanced through reducing inherent power imbalances and increasing self-determination.

## 4. Conclusions

Indigenous cultures as a determinant of health continue to be incorporated into public policy instruments, like the Health Plan’s second Implementation Plan. The integration of cultural beliefs and practices into healthcare services is therefore warranted. Importantly, culture’s innate relationship to Indigenous knowledges of ways of being, knowing, and doing must be understood. However, conceptual and contextual difficulties remain. Nevertheless, lessons from case examples like the Katherine Hospital and the stories from research participants have provided learning opportunities. Similarly, the new Aboriginal and Torres Strait Islander six actions contained in the 2nd Edition of the National Safety and Quality in Health Service Standards provide healthcare providers with governance and accountability examples of how to embed Indigenous cultures and Indigenous knowledges into the operations of all healthcare services.

## Figures and Tables

**Figure 1 ijerph-21-00003-f001:**
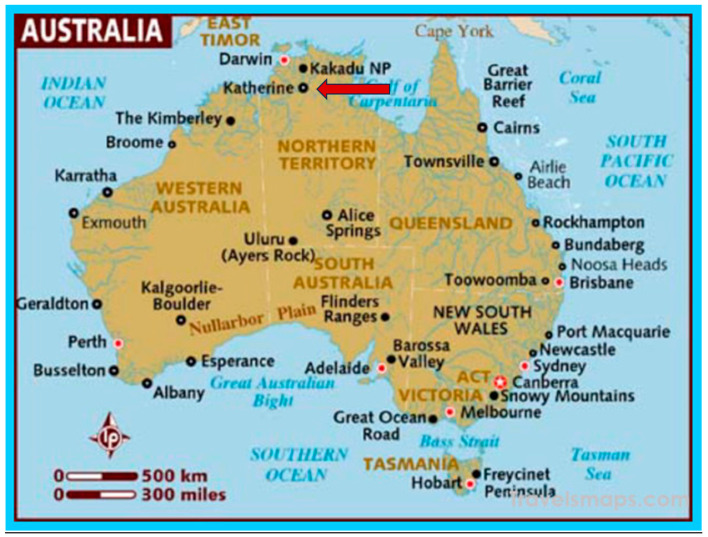
Map of Australia showing the location of Katherine (see the red arrow) and Darwin. Source: Creative Commons: Maps of Australia, Travelmaps.com http://travelsmaps.com/map-of-australia.html. Accessed on 29 October 2023.

**Figure 2 ijerph-21-00003-f002:**
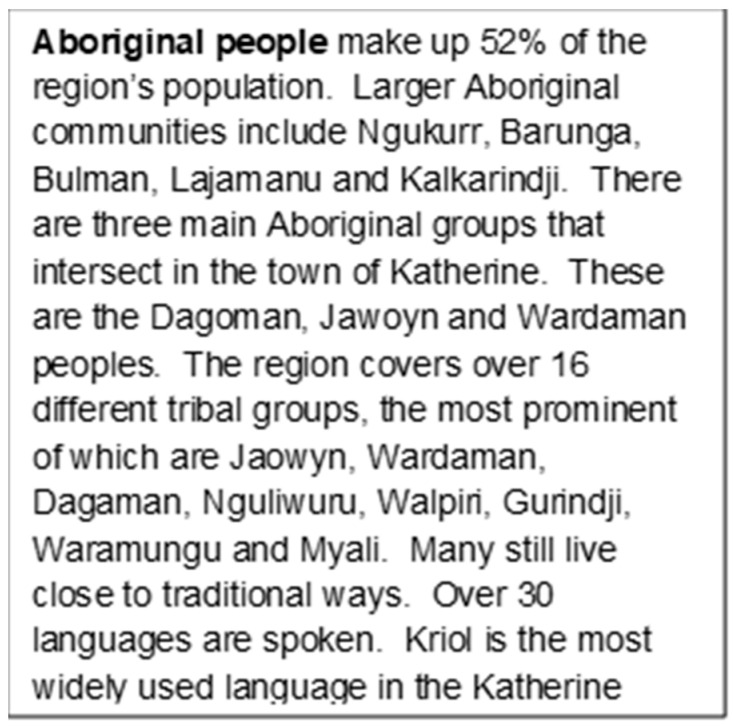
Description of the Aboriginal population within the Katherine Hospital catchment area. Source: Royal Australasian College of Physicians. *Building an outreach program in remote Indigenous communities Sydney: Royal Australasian College of Physicians*; 2022. Available from: https://www.racp.edu.au/advocacy/policy-and-advocacy-priorities/medical-specialist-access-framework/medical-specialist-access-framework-case-studies/building-an-outreach-program-in-remote-indigenous-communities. Accessed on 29 October 2023.

## Data Availability

Restrictions apply to the availability of these data. Data was obtained from research participants and cannot be shared due to privacy and confidential agreements.

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
