# Peer review of "Putting Indigenous Cultures and Indigenous Knowledges Front and Centre to Clinical Practice: Katherine Hospital Case Example"

_ijerph, 2023, doi:10.3390/ijerph21010003_

Round 1

Reviewer 1 Report

Comments and Suggestions for Authors

see attached

Comments on the Quality of English Language

Reviewer 2 Report

Comments and Suggestions for Authors

Comments on the Quality of English Language

Round 2

Reviewer 1 Report

Comments and Suggestions for Authors

The authors' response together with a read of the revised manuscript convinces me this paper is ready for publication and will make a meaningful contribution to the literature.

Author Response

Thank you for your review of our paper and for approving it for publication on the 2nd round

Reviewer 2 Report

Comments and Suggestions for Authors

·      Who are “policy actors?” Is this the same as a policy stakeholder, or something different?

·      Typo: Actor’s  actors’

·      The concept of “Country” as meant in for the Indigenous context is first mentioned at section 1.2, and I think this is where the definition needs to be. The definition provided a bit later, is very clear and helpful; I just think it should come sooner. Also, the manuscript is not consistent with either using “Country” or “country.” I assume “Country” is more accurate, but either way, I recommend consistency.

·      Can you say more about this radio program? Is it a series? A one-off program? Is it sponsored by the hospital? What is its objective?

·      Also, can you provide more details about the Katherine Hospital? How big is it? Is it a tertiary hospital? Is it the only full-service hospital in the area? It would help to understand the significance of the changes they have made in their care to know more about the relative importance of this hospital to the community.

·      I’m still confused as to when quotes are representing the current study’s dataset, quotes from the radio program, and quotes from previous studies. Is it possible to perhaps add a table that describes the 2018 dataset (e.g., number of interviewees, basic demographics (age, gender), and their role in policy or healthcare)?

·      Can the questions or objectives of this study be more clearly stated? I see this statement early in the introduction as the objective of the study statement:

o   Drawing on new evidence from policy actors’ yarns, we use their stories to assist in our learnings of how healthcare providers and services, can support the implementation of the cultural determinants of health, permanently into organizational and clinical prac-tice.

However, I think this is a good place to say clearly: 1) who the policy actors are; 2) state what yarns/yarning are/is (I had to look it up to understand what the method was); 3) what are their stories responding to (just the radio program? A series of prompts?)? More details of the program or prompts can come later, like in a methods section, but this would help set up this research better for the reader.

·      The additional paragraph at the end of the “allies and accomplices” section helps tie that perspective back to the previous sections that reference people’s experiences receiving or learning to give culturally resonant healthcare.

·      Why not just say: “The cultural determinants of health are core to the National Aboriginal Health Plan,” rather than:

o   “Culture has been exhibited as core to the National Aboriginal Health plan – known as the cultural determinants of health.”

Comments on the Quality of English Language

The English language is fine, but could still use some copy editing. Quotations are inconsistently punctuated, and still some minor errors (e.g., actor's should be actors').

Author Response

Review Report Form

2nd reviewer’s feedback and comments

(x) I would not like to sign my review report

( ) I would like to sign my review report

Quality of English Language

( ) I am not qualified to assess the quality of English in this paper

( ) English very difficult to understand/incomprehensible

( ) Extensive editing of English language required

( ) Moderate editing of English language required

(x) Minor editing of English language required

( ) English language fine. No issues detected

Is the work a significant contribution to the field?             

Is the work well organized and comprehensively described?         

Is the work scientifically sound and not misleading?         

Are there appropriate and adequate references to related and previous work?    

Is the English used correct and readable?              

Comments and Suggestions for Authors

  • Who are “policy actors?” Is this the same as a policy stakeholder, or something different?

To minimise confusion, we have replaced policy actors with research participants mainly because in an international arena, policy actors would have many meanings.

      Typo: Actor’s  actors’

We have corrected this. Thanks for pointing this out

  • The concept of “Country” as meant in for the Indigenous context is first mentioned at section 1.2, and I think this is where the definition needs to be. The definition provided a bit later, is very clear and helpful; I just think it should come sooner. Also, the manuscript is not consistent with either using “Country” or “country.” I assume “Country” is more accurate, but either way, I recommend consistency.

Thank you for alerting us to this inconsistency. We have now corrected and made all Country. We have also moved the definition of Country up, so it now sits under section 1.2. Thank you because this has improved flow and readability

  • Can you say more about this radio program? Is it a series? A one-off program? Is it sponsored by the hospital? What is its objective?

We have already clarified this in the text, at the top of page 2, where we have stated that it is a program on the National Broadcaster in Australia, the ABC (a bit like the BBC in Britain), and this broadcast was an episode of the weekly, nationally broadcast program Background Briefing. It has nothing to do with the Katherine Hospital and is a national radio program highly accessed across Australia. However, based on this feedback, we have added a sentence aimed at making this clearer.

Background Briefly is a weekly program exploring a diversity of social and political issues from across Australia and is broadcast nationally on the ABC, which is Australia’s government-funded radio and television national broadcasting network.

  • Also, can you provide more details about the Katherine Hospital? How big is it? Is it a tertiary hospital? Is it the only full-service hospital in the area? It would help to understand the significance of the changes they have made in their care to know more about the relative importance of this hospital to the community.

We have added this just under 2.1 section

Katherine Hospital is a district public hospital servicing the Katherine Region in the NT. It services an area of 336,674 km²., with 85% percent of its patients being Aboriginal people, many from some of the most remote communities in Australia. It is operated by the NT  Government Department of Health (Northern Territory Government, 2023)

  • I’m still confused as to when quotes are representing the current study’s dataset, quotes from the radio program, and quotes from previous studies. Is it possible to perhaps add a table that describes the 2018 dataset (e.g., number of interviewees, basic demographics (age, gender), and their role in policy or healthcare)?

We have addressed the reviewer's above confusion in the following way.

  1. Explained and expanded on the methodology and methods used in this paper in relation to the use of Indigenous research methodologies.
  2. Wrote more about storying, including yarning and storytelling, to reiterate that we also apply this method in the manuscript.

Overall, though, we make the following remark as well.

This paper is unique because we present the material that honours Indigenous sciences and knowledges of cultural ways of being, knowing and doing, which decolonises the dominance of non-Indigenous Western sciences.

The confusion being experienced could be due to the conflict that is often faced when one’s positionality and ontological, epistemological and auxological standpoint are in tension with others, especially when Indigenous sciences, which is a valid area of scholarship, is being applied. In this paper, we use storying as a methodological and rigorous way of presenting a story told in a study and drawn from that study. A form of conversational style story and vignettes that provide and confirm the story of how Katherine Hospital has emerged from being one of the worst examples of culturally safe care to one of the best. We authenticate that this style of Indigenous writing and publishing ‘data’ is fitting and continues to decolonise the dominant research paradigms of Western bio-medical sciences. Therefore, putting these into a table would fracture and Westernise this and detract from the Indigenous ways of ‘doing’ research and publishing. So, no table will be included.

Also, we added more spacing between quotes to make this clearer and smoother for the reader; however, the journal has now removed all that extra space.

  • Can the questions or objectives of this study be more clearly stated? I see this statement early in the introduction as the objective of the study statement:

o   Drawing on new evidence from policy actors’ yarns, we use their stories to assist in our learnings of how healthcare providers and services, can support the implementation of the cultural determinants of health, permanently into organizational and clinical prac-tice.

The intent of this paper is NOT about the study, such as the research questions, objectives, or findings. The findings of this study are disseminated elsewhere. Rather, this paper is designed to showcase the Katherine Hospital for turning their practice from one of the most culturally unsafe to showing the way to other hospitals struggling with ‘taking their own leave’. We focus on contributing to the knowledge base of implementing the cultural determinants of health and have further strengthened such an approach. Similarly, we deliberately refrain from providing too much to the study rationale and findings because that is not the aim of this paper, which is about contributing to a practice evidence gap on implementing the cultural determinants of health. Therefore, the material provided about the study in the manuscript is sufficient.

However, I think this is a good place to say clearly: 1) who the policy actors are; 2) state what yarns/yarning are/is (I had to look it up to understand what the method was); 3) what are their stories responding to (just the radio program? A series of prompts?)? More details of the program or prompts can come later, like in a methods section, but this would help set up this research better for the reader.

Thanks for the reviewer's comments here. We have replaced policy actors with research participants to avoid having a clearer meaning of policy actors, which is contestable in an international arena. Further, we have made it clearer what yarns and yarning are within an Indigenous research methodology, especially the use of storying and storytelling that weaves lessons we are learning with the 2020 earlier study research participant stories. We reaffirm, as mentioned above, that this is not a research paper aimed to report on its methods or findings. Furthermore, this paper is unique in that we are decolonising Western scientific ways of how to present knowledge and the way that knowledge is disseminated. In this case, we are disseminating scant and new knowledge about implementing the cultural determinants of health.

  • The additional paragraph at the end of the “allies and accomplices” section helps tie that perspective back to the previous sections that reference people’s experiences receiving or learning to give culturally resonant healthcare.

Thank you

  • Why not just say: “The cultural determinants of health are core to the National Aboriginal Health Plan,” rather than:

o   “Culture has been exhibited as core to the National Aboriginal Health plan – known as the cultural determinants of health.”

We would like to retain this statement because this is what is stated in that document, and it is a key statement we wish to open the discussion with

Comments on the Quality of English Language

The English language is fine, but could still use some copy editing. Quotations are inconsistently punctuated, and still some minor errors (e.g., actor's should be actors').

We have checked and rechecked now for minor errors and done some corrections within the paper